# Vitamins in Alzheimer’s Disease—Review of the Latest Reports

**DOI:** 10.3390/nu12113458

**Published:** 2020-11-11

**Authors:** Anita Mielech, Anna Puścion-Jakubik, Renata Markiewicz-Żukowska, Katarzyna Socha

**Affiliations:** Department of Bromatology, Faculty of Pharmacy with the Division of Laboratory Medicine, Medical University of Białystok, Mickiewicza 2D Street, 15-222 Białystok, Poland; an.mielech@gmail.com (A.M.); renmar@poczta.onet.pl (R.M.-Ż.); katarzyna.socha@umb.edu.pl (K.S.)

**Keywords:** Alzheimer disease, nutrition, antioxidant

## Abstract

Alzheimer’s disease (AD) is the most common form of dementia, and the aging of the population means that the number of cases is successively increasing. The cause of the disease has not been established, but it is suggested that many factors affect it, including nutritional aspects. As part of the work, the PubMed database has been searched, beginning from 2005, for terms related to key nutritional aspects. A diet rich in antioxidant vitamins can improve the cognitive functions of patients. Thanks to an adequate intake of B vitamins, homocysteine levels are reduced, which indirectly protects against the development of the disease. A properly balanced diet, as well as the use of appropriate supplementation, can contribute to improving the clinical condition of patients with AD.

## 1. Introduction

Alzheimer’s disease (AD) is one of the most common forms of dementia in the world, accounting for 2/3 of all dementias. In aging societies, the number of AD patients is predicted to increase. For example, in 2005 in Poland there were 5.9 million people over 60, and according to estimates, by 2030 this number will have increased to over 9 million [1,2]. In 2050, the disease will have affected about 115 million people worldwide [3].

The cause of AD is still unknown, but there are several hypotheses that attempt to explain the etiology of the disease. Usually, the cause is multifactorial. The gold standard in diagnosis is the accumulation of β-amyloid plaques inside the cerebral cortex and the presence of tangles of intra-nerve filamentous neurons containing tau proteins, as well as amyloid angiopathy [4]. The role of microRNA (miRNA) in neuronal cell death that is induced by β amyloid is emphasized [5]. About 95% of the cases begin at the age of 60–65 [4].

In AD, memory disorders and progressive cognitive dysfunctions occur, such as speech disorders, impairment of visual-spatial skills, and behavioral disorders [6,7]. Many symptoms, such as sleep disturbances and deterioration of mood and anxiety are revealed before clinical diagnosis of dementia. Symptoms such as confusion, aggressive behavior, and delusions may remain undiagnosed for a long time [8]. In AD, apart from the classic amnestic form, other forms are distinguished, with predominant language disorders (referred to as the speech or language variant), the visual variant, the variant with behavior disorders (frontal variant), and the form with apraxia (apractic variant) (Table 1) [9].

Currently, AD therapy uses drugs from the group of acetylcholinesterase inhibitors, such as rivastigmine and donepezil, slowing down the breakdown of acetylcholine, as well as memantine, an N-methyl-D-aspartate receptor (NMDA receptor) antagonist, whose task is to block glutamate receptors. Memantine is recommended for moderate and severe patients. Unfortunately, none of the current pharmacological treatments can stop the disease, but only decelerate its progression [10].

There are reports of common pathophysiology of two amyloidoses, AD and type 2 diabetes. In both conditions, insulin resistance, carbohydrate disorders, systemic oxidative stress, inflammation, amyloid accumulation, and cognitive decline occur. These features may indicate a common basis for these diseases. AD with concomitant type 2 diabetes is associated with so-called brain insulin resistance, which may increase the additional pathophysiological mechanisms of AD. The onset of type 2 diabetes increases the risk of AD by 50 to 100%, which is why a well-balanced diet can delay this process [11].

Optimal supply of adequate nutrients (e.g., by introducing a Mediterranean diet) or increased consumption of selected ingredients (for example, a ketogenic diet) [12], as well as avoiding toxic elements, are essential for the proper functioning of the brain and, consequently, the prevention of neurodegenerative diseases. Therefore, this publication reviews the latest literature on supporting AD therapy by taking vitamins, mainly with antioxidant properties.

## 2. Materials and Methods

This publication reviewed research on the importance of nutrition in AD, published from 2005 to 2020. Only human models were included in the review, with a minimum of 100 participants. In the case of several basic research papers and textbooks, reference was made to the older studies. The PubMed publication database searched for “Alzheimer’s disease”, “cognitive impairment”, “diet”, “vitamins”, “intake”, “level in serum”. The full texts of selected papers were analyzed in order to assess whether they met the adopted criteria. The publications available in a language other than English were excluded. This review presents the most important publications describing the state of the art regarding vitamin intake in AD.

## 3. Results and Discussion

### 3.1. B Vitamins

Vitamins B are a diverse group in terms of structure and function. Folates, which provide methyl groups necessary for DNA methylation, play an important role in the pathogenesis of neurological diseases [13]. The beneficial effect of vitamin B on cognitive function is associated with the metabolism of homocysteine, which contributes to increased cardiovascular risk and increased cognitive impairment [14]. In patients with AD, there is a decrease in DNA methylation. Nutrition can affect the regulation of gene expression and, thus, indirectly, can have an impact on the course of the disease. These vitamins are substances closely related to the methionine cycle [7]. Vitamin B12 is involved in the transformation of homocysteine to methionine, and vitamin B6 and folic acid are cofactors for this reaction. The deficiency of the above vitamins causes accumulation of homocysteine in the body [15]. This reduces the concentration of S-adenosylmethionine, which is a methyl donor. DNA demethylation is induced, which increases the overexpression of genes associated with the occurrence of AD [7].

Table 2 contains data on the intervention and measurement of B vitamins in AD patients.

The effect of B group vitamins (folic acid, pyridoxine and cobalamin) on homocysteine levels was confirmed in 266 participants over 70 years of age with mild cognitive impairment (MCI). Half of the participants (*n* = 133) were in the placebo group. The following supplementation was used: 0.8 mg folic acid, 20 mg vitamin B6, and 0.5 mg vitamin B12 for 2 years. As a result of therapy with group B vitamins, homocysteine concentration in the supplemented group was 30% lower compared to the placebo group. It was noted that patients undergoing supplementation (initial methionine concentration above 11.3 μmol/L) achieved better results in the Mini-Mental State Examination (MMSE) test, episodic memory, and semantic memory [16].

In a randomized study conducted in newly diagnosed AD patients over 60 years of age (*n* = 121), the effect of supplementation with folic acid on the inflammatory process and on cognition was assessed. Patients used donepezil, and for half a year they were additionally supplemented with folic acid at a dose of 1.25 mg/day. Supplemented patients obtained statistically higher results in the MMSE test compared to the control group, which did not receive folic acid. The authors emphasize that inflammation may play an important role in the interaction between AD and folic acid [17].

Different effects were noticed in other studies, where supplementation with vitamin B12 and folic acid did not reduce the development of cognitive disorders in elderly people with elevated levels of homocysteine; homocysteine concentration after 24 months of supplementation (9.3 ± 2.4 μmol/L), folate (48.0 ± 12.6 nmol/L), and active vitamin B12 (123.6 ± 13.6 pmol/L) [18].

A multicenter, double-blind study involving 2919 patients over 65 years of age, studied the effects of ingestion of 400 μg of folic acid and 500 μg of vitamin B12. The placebo and study group received an additional 15 μg of vitamin D3. The authors did not show a beneficial effect on cognitive skills. However, they emphasized that the above supplementation might slow down cognitive impairment [19].

In the study conducted by Oulhaj et al. [20], 266 participants with mild cognitive impairment were divided into a supplement group (folic acid, vitamin B6 and B12) for 2 years and a placebo group. It was shown that treatment with group B vitamins did not slow down the cognitive dysfunctions when the concentration of omega-3 fatty acids was low, whereas when the concentration was above 579 μmol/L, these vitamins inhibited the development of dementia processes, which was confirmed by Hopkins Verbal Learning Test-Revised (HVLT-DR) longitudinal scores of episodic memory. Treatment with vitamin B caused a decrease in total homocysteine levels, which resulted in a reduction in the rate of gray matter atrophy.

A cross-sectional study conducted by Meng et al. [21] on 4605 participants demonstrated that higher serum homocysteine levels were associated with an increased risk of AD. High levels of folic acid and vitamin B12 were factors protecting against cognitive impairment [21].

Ulusu et al. [22] attempted to use the determination of vitamin B12 and folic acid in a group of 290 patients as a marker of dementia. This study needs to be reconducted on a larger number of patients because a classification error was made: 5 patients had normal serum folate levels and the reference range was 4.6 to 18.7 ng/mL.

A cohort study by An et al., published in 2019 [23] involving 2533 participants with dementia or normal cognitive function, found that insufficient vitamin B12 intake contributed to cognitive decline, and adequate levels of folic acid, vitamin B6, and vitamin B12 had a beneficial effect on the cognitive resources of participants without cognitive impairment.

In summary, the effect of nutritional intervention in the form of dietary supplementation with vitamins B seems to be inconclusive, as shown by the analysis of individual publications. While it has been indicated that the measurement of serum folate and vitamin B12 levels is not a reliable test, it is suggested that high serum levels of vitamin B12 may be a protective factor.

Recommendations for the consumption of vitamin B for adults have been set at the level of the estimated average requirement (EAR) for B1, 1.1 and 1.3 mg of thiamine (for men and women, respectively); for B2, 1.1 and 0.9 mg of riboflavin; for B3, 12 and 11 mg niacin equivalent; for B6, 1.4 and 1.3 mg of vitamin B6, 320 µg of folate equivalent and 2 µg of cobalamin for both gender [24].

### 3.2. Antioxidant Vitamins

Increased oxidative stress is one of the factors contributing to the onset and advancement of neurodegenerative processes. Impaired antioxidant processes, excessive free radical accumulation, and age-related oxidative stress of brain tissue and lymphocytes are noted [25]. Pleiotropic mechanism of antioxidants, manifested, e.g., in a reduction of oxidized lipid membranes, reduction of nucleic acid damage or prevention of carbonylation of proteins, prevents the identification of a damaged specific pathway that is responsible for the formation of AD [26].

In a 2014 meta-analysis, levels of vitamins, microelements, and fatty acids in the blood were assessed in patients with AD and in a group of elderly people without cognitive impairment. Statistically significant vitamin deficiencies were found in A (9 tests), C (8 tests), E (20 tests), folates (31 tests), and vitamin B12 (32 tests) in the group of patients with AD, which indicates the need to provide the correct amounts of the above ingredients in diet or, in the absence of such a possibility, through supplementation. It was emphasized that AD patients have impaired systemic availability of the studied nutrients [27].

#### 3.2.1. Vitamin A

A characteristic feature of inflammation of the nervous system is microglial activation. Inflammation occurs in chronic and acute neuropsychiatric diseases. Literature data indicates that microglia activation is known to be one of the causes of AD. Moreover, malfunctioning of the microglia may result in changes in local concentrations of retinoic acid [28]. In addition, vitamin A and its derivatives play an important role in the differentiation of nerve cells, as well as in the expression of neurotransmitters in the brain and gene expression through interaction with retinoic acid and retinoid X receptors [29].

A literature review showed that vitamin A was the least studied of the antioxidant vitamins. Vitamin A intake or dietary intake may improve cognitive function in AD patients, as shown in Table 3.

Low dietary intake of vitamin A is associated with an increased risk of dementia, which was confirmed in the group of 333 participants [30].

A study by Yuan et al. (2020) showed that higher consumption of β-carotene reduced the risk of cognitive impairment [31]. There are also studies that do not confirm the beneficial effects of this ingredient.

In a prospective study conducted by Pèneau et al., it was shown that the consumption of fruits, which are a source of β-carotene, did not reduce the risk of cognitive dysfunction [32]. Other studies on the effects of vitamin A on AD do not meet the inclusion criteria for this article.

Moreover, vitamin A is involved in neuronal differentiation and also influences the secretion of neurotransmitters in the brain [29,33].

In conclusion, it has been shown that AD patients are deficient in vitamin A, but in the case of increased consumption of β-carotene, the assessment of its effect on cognitive abilities has failed to provide definitive evidence of benefit. For example, the vitamin A intake recommendations for the population in Poland, at the level of the EAR is 630 μg of retinol equivalent/day for men, for women: 500 μg/day [24].

#### 3.2.2. Vitamin C and E

Vitamin E can be stored in the central nervous system, where it reduces lipid peroxidation and β-amyloid deposition. Therapy with antioxidant substances should be implemented in the earliest possible stage of the disease [34].

In patients with AD, there is a correlation between blood vitamin C and vitamin E levels and dementia [35]. Ascorbic acid has a neuroprotective effect because it has the ability to scavenge free radicals, reduces β-amyloid activity, and is also involved in the chelation of iron, zinc, and copper. It is a key antioxidant of the central nervous system [36]. In a rat study, a pro-oxidative diet has been shown to increase the level of amyloid precursor protein [37].

Table 4 summarizes the data on the effect of vitamin C and E intake or supplementation on the cognitive abilities of AD patients.

In the study conducted on 613 patients, the effects of α-tocopherol (at a dose of 2000 IU/day), memantine (20 mg/day), and combinations of these substances were assessed. The observation period was 2.27 years. Vitamin E alone proved to slow down cognitive decline in patients with mild to moderate AD. Interestingly, memantine or memantine therapy, together with α-tocopherol gave less beneficial effects [38].

The results of studies on vitamin E supplementation in patients with AD are inconclusive. In a cohort study, 7540 patients over 60 years of age were treated with long-term supplementation with vitamin E (400 IU/day), selenium (200 μg/day), and vitamin E with selenium (duration of use: 5.4 ± 1.2 years), but the therapy did not prevent dementia [39]. Similar observations were obtained in a study assessing vitamin E supplementation at a dose of 300 mg and 400 mg of vitamin C on the cognitive function of the elderly with MCI (*n* = 256, age: 60–75 years). The intervention lasted a year and resulted in a reduction of the oxidative stress of the body, including a decrease in the level of malonic aldehyde, but did not improve cognitive abilities [40].

Higher brain γ-tocopherol levels are associated with presynaptic protein levels in the midfrontal cortex. These results suggest that vitamin E plays a significant role in maintaining presynaptic protein levels (synaptophysin, synaptotagmin, septin 5, syntaxin, synaptosomal associated protein 25 (SNAP-25), vesicle-associated membrane protein (VAMP), complexin-I, complexin-II) [41].

Patients with mild cognitive impairment (MCI, *n* = 166) and people diagnosed with AD (*n* = 168) have been shown to have significantly lower total tocopherols (6.80 μmol/mmol cholesterol and 6.40 μmol/mmol, respectively), total tocotrienols (97.28 nM/mmol cholesterol and 91.33 nM/mmol cholesterol, respectively) and total vitamin E in the blood (6.9 μmol/mmol cholesterol and 6.49 μmol/mmol), compared to persons with normal cognitive abilities (CN, *n* = 187), whose total tocopherols were 7.67 μmol/mmol cholesterol, total tocotrienols—118.02 nM/mmol cholesterol, total vitamin E—7.8 μmol/mmol cholesterol [42].

In a 13-year study on a group of 2533 French people aged 45 to 60, the relationship between fruit and vegetable consumption (400 g) and cognitive ability was assessed using neuropsychological tests. It was proved that the consumption of fruit and other products rich in vitamin C and vitamin E positively correlated with the results of verbal memory test, in contrast to the consumption of vegetables and vegetables rich in β-carotene [32].

Another prospective, longitudinal cohort study (*n* = 9250) also confirmed the beneficial effect of fruit consumption on reducing the risk of AD [43].

A review of the literature from 2012 assessing the effect of ascorbic acid intake on cognitive functions revealed ambiguous information about the effectiveness of the use of vitamin C in AD. The hypothesis was presented that the brain draws ascorbic acid from the peripheral pool to endure oxidative stress [44].

In conclusion, research results indicate that decreased vitamin E concentration may be associated with an increased risk of AD, supplementation at higher doses (2000 IU) than lower doses (400 IU) is more effective, and increased consumption of antioxidant vitamins (E and C) may help to improve cognitive abilities.

Vitamin C intake by adults should be 75 mg/day for men and 60 mg/day for women. Standards are set at the EAR level. The vitamin E intake recommendations at the level of the adequate intake (AI) is 10 μg of retinol equivalent/day for men, for women, 8 μg/day [24].

#### 3.2.3. Vitamin D

Vitamin D also plays an important role in neurodegenerative processes. Its deficiency is a genetic risk factor for AD, Parkinson’s disease, multiple sclerosis, and vascular dementia [45].

Table 5 summarizes the research on the effects of vitamin D on the cognitive abilities of patients with AD.

In randomized, double-blinded placebo-controlled study on 2044 participants, supplementation of 1000 mg of calcium carbonate with 400 IU of vitamin D(3) did not improve cognitive impairment [46].

The effect of vitamin D levels on cognitive functions was studied in a group of 146 patients (78 women and 68 men). Among other things, brain imaging by magnetic resonance imaging was performed. It was shown that patients who obtained lower results in the MMSE test were characterized by lower serum vitamin D concentration [47].

Low blood levels of vitamin D may increase cognitive decline in people with dementia [48,49]. The InCHIANTI study, conducted on 858 patients over 65 years of age, showed that low serum 25-hydroxyvitamin levels were associated with an increased risk of cognitive impairment. When assessing the vitamin’s concentration in patients with a deficiency (below 25 nmol/L) and patients with normal levels (above 75 nmol/L), the relative risk of corrected cognitive decline was 1.6 [49].

In two cohort studies, no protective effect of high vitamin D levels on cognitive performance was observed [50,51].

Leeuw et al. [52] evaluated 28 nutritional biomarkers in blood and 5 in cerebrospinal fluid. It was shown that higher levels of w1.25 (OH) in serum were associated with impaired cognitive functions. Among other factors, the authors also mentioned low levels of S-adenosylmethionine and high levels of HDL, theobromine, cholesterol, and iron.

Vitamin D has the ability to reduce inflammation in the hippocampus and mitigate the accumulation of β-amyloid in the process of increased phagocytosis. This vitamin stabilizes the activity of calcium channels, regulating calcium homeostasis, disturbed by the deposition of β-amyloid. In addition, it increases the expression of vitamin D receptor, which further exerts an antioxidant effect [53]. It has been shown that vitamin D therapy in the course of AD prevents the synthesis of β-amyloid and promotes the clearance of this peptide from the brain [45]. A dose of 50,000 IU, used 3 times a week for 4 weeks, was safe and effectively normalized 25(OH) D levels in the blood serum of patients [54]. In a Annweiler et al. protocol (2011) [55], it was assumed that supplementation with 100,000 IU of cholecalciferol with memantine for one month enhanced cognition and memory capacity in patients with moderate AD. There is evidence that high vitamin D levels improve cognition, particularly in mild AD [56].

In addition, it is suspected that the use of vitamin D together with memantine in patients with moderate forms of the disease improves cognitive performance and memory ability by reducing the loss of neurons. The neuroprotective effects of vitamin D and memantine may be potentiated [53].

Summing up, a low level of vitamin D in the serum is associated with the risk of cognitive disorders, but supplementation with a dose of 400 IU does not bring any results (elderly people are recommended several times higher doses).

For example, the vitamin D intake recommendations for the population in Poland, at the level of the AI is 15 μg of cholecalciferol/day for men and for women [24].

#### 3.2.4. Other Factors

Recently published systematic reviews and meta-analyses show the beneficial effects of various diets, including the Mediterranean diet and the “healthy diet”, which are based, among other things, on consuming more foods that provide the vitamins discussed in this study.

There is no correlation between meat consumption and the occurrence of cognitive impairment [57].

The Mediterranean diet is characterized by high consumption of vegetables and fruits, which are exceptionally rich in water-soluble vitamins. It has been shown that the Mediterranean diet protects against the development of cognitive impairment [58], whereas a ketogenic diet may have an impact on memory functions due to its influence on neurotransmission. In a systemic review by Grammatikopoulou et al., the application of ketogenic therapy (Medium Chain Triglicerydes, MCT supplementation), both long-term and interventional, improved cognitive functions [59].

The effect of “healthy diets” on cognition has also been studied. It has been reported that healthy eating patterns reduce the risk of dementia, while a high-fat diet and diets with a high glycemic index increase neuronal damage [60,61]. Proinflammatory diet also may increase the progression of dementia [62].

It has been proven that the use of nutritional formulas, fatty acids, ginseng, inositol, probiotics, and products rich in flavonoids may delay the progression of dementia [63,64]. On the other hand, according to Heider et al.’s (2020) systematic review and meta-analyses, supplementation of vitamin B, E, omega-3, and nutritional formulas does not improve cognitive function [65].

Table 6 summarizes the meta-analyzes in which the impact of consumption of B vitamins, vitamins C, D, and E on the health of AD patients was assessed.

In summary, the impact of most of the vitamins on cognitive function remains ambiguous. As far as B vitamins are concerned, the reasons authors give for the inefficiency of supplementation include: the insufficient size of the group to detect differences in cognitive abilities, the load on blood vessels, and the use of other drugs (which could reduce patients’ sensitivity to supplementation), no imaging of brain lesions, which is the most sensitive and reliable method of assessing changes in the brain [18], the interaction between, among others, vitamin B12 and vitamin D, which could weaken the effect of vitamin B12 [19]; in the case of the absence of expected effects of increased intake or supplementation with vitamin A (which is a fat-soluble vitamin): increased fatty acid content in patients, possibly body fat [30]; in the case of no improvement in cognitive function after an intervention involving supplementation with vitamin E, the authors point to inappropriate group selection (e.g., only one sex), too short duration of supplementation, too low doses of vitamin [39]. As for the reasons for the lack of effectiveness of vitamin D supplementation, the authors emphasize, inter alia, taking only 1 measurement, different levels of education, and groups disparate in terms of age [48].

Future research on people with Alzheimer’s disease should focus on further evaluation of the impact of vitamin intake on cognitive performance in patients in order to be able to clearly define their effect and formulate international guidelines for diet and supplementation in patients with AD. In addition, they should explain the mechanisms that are not fully understood to be able to develop appropriate supplement formulations that would be easily distributed in the patient’s body.

To promote healthy aging, governments in many countries are introducing public health support programs. For example, the Universities of the Third Age are established that support the memory and cognitive abilities of the elderly. Older people, for economic reasons, do not use all the medications recommended for their specific conditions, therefore, programs are being introduced under which seniors receive some drugs for free (in Poland, it is the “Drugs75+” program). In addition, pharmaceutical care (including care for the elderly) is developing in Europe, the aim of which is to help senior citizens in proper pharmacotherapy: assessing whether the preparations and dietary supplements used do not interact, and whether the elderly person correctly uses, e.g., an inhaler or a glucometer (people with AD may have difficulty with this). Containers are sold in which the AD caregiver can distribute medications for the entire week. In addition, insufficient intake of vitamins may be related to the general malnutrition of the elderly. There are programs in Europe to prevent malnutrition in older people, this problem affects 13.5 to 29.7% of the elderly, including PROMISS (Prevention of Malnutrition In Senior Subjects in the EU) [74].

## 4. Conclusions

Research suggests that a diet rich in antioxidant vitamins may slow the progression of Alzheimer’s disease. Data on supplementation with vitamins A, B, C, D, and E vitamins are ambiguous. There are no international guidelines for dietary recommendations in AD. The conducted meta-analyses suggest that vitamins A, B, C, D, and E deficiencies may increase the risk of AD, therefore, a diet rich in vitamins A, D, C, B6, B12, and folic acid, as well as supplementation with higher doses (e.g., 2000 IE of vitamin E) seems to be recommended in the prevention of Alzheimer’s disease. Adequate quantities of fruit and vegetables will supplement the deficiency of antioxidant vitamins, which are often deficient in patients with dementia. Further research should focus on resolving unclear issues.

## Figures and Tables

**Table 1 nutrients-12-03458-t001:** Characteristics of symptoms of different variants of Alzheimer’s disease [9].

Language Variant/Logopenic Progressive Aphasia (LPA)	Frontal Variant	Visual Variant/Posterior Cortical Atrophy (PCA)	Apraxic Variant
For at least the first two years, there is only an isolated language deficit of unnoticeable onset, progressive, without previous language disorders, and no other specific causes for the above disturbances.	Pronounced disturbances in behavior and executive functions. These often mimic the behavioral disturbances in frontotemporal dementia.	Initially, visual-spatial disorders dominate without any evident ophthalmic pathology, later psychotic symptoms appear, and in the final stage, there is a dementia syndrome with deep cognitive deficits, disturbing everyday functioning.	Inability or difficulty in performing learned movements, while understanding the command, the willingness to perform it. The lack of impaired motor coordination, and weakened sensation and muscle strength.
After two years, other cognitive functions deteriorate, although the language deficit dominates and deepens much faster than the other ones.		This is followed by the inability to see more than one object or one element of an object at the same time (simultaneous diagnosis), finger agnosia, page orientation disorders (right/left), and apraxia.	

**Table 2 nutrients-12-03458-t002:** Summary of studies on vitamin B intake and measurements in AD.

Vitamins	Type of Study	Number of Participants/Age [Year]	Assessment	Intervention/ Measurements	Duration	Results	References
INTERVENTIONS							
Vitamins B	randomized controlled trial	266/≥70	MMSE, Hopkins Verbal Learning Test, CVLT	0.8 mg folic acid + 20 mg vitamin B6 + 0.5 mg vitamin B12/day	2 years	(+) slowing cognitive decline	[16]
Folic acid	randomized controlled trial	121/40–90	MMSE, ADL	1.25 mg of folic acid + donepezil/day	6 months	(+) beneficial effects in patients with AD	[17]
Vitamins B	randomized placebo controlled trial	279/≥65	MMSE, NTB, CDR	500 μg methylcobalamin + 400 μg folic acid or placebo/day	2 years	(-) lack of reduction of cognitive decline	[18]
Vitamin B12, folic acid	randomized, placebo-controlled trial	2919/74.1 ± 6.5	MMSE and others tests	400 μg of folic acid + 500 μg of vitamin B12/day	2 years	(-) lack of reduction of cognitive decline	[19]
MEASUREMENTS							
Folic acid, vitamins B6 and B12	randomized trial	266/≥70	CDR	folic acid, vitamins B6 and B12 measurements in serum	2 years	(+) the effectiveness of vitamin B therapy when omega-3 levels are in the upper normal range	[20]
B12	cross-sectional study	4605 (2396 with AD)/≥55	China’s validation MMSE	homocysteine and B12 serum levels	1 measurement	(+) high folate and B12 levels are protective factors	[21]
Vitamin B12 and folic acid	cross-sectional study	290/geriatric patients	The authors performed tests	vitamin B12 and folic acid level	1 measurement	(−) serum folic and vitamin B12 levels are not reliable tests for screening presymptomatic AD	[22]
CONSUMPTION							
Vitamins B	cohort study	2533/50–70	-	diet intake of vitamin B12, B6	2.3 years	(−) low intake of vitamin B12 intensify cognitive decline	[23]

(+)—confirmed or (−)—no confirmed association of intervention/concentration measurement/consumption with Alzheimer’s disease, ADL, Activities of Daily Living scale; CDR, Clinical Dementia Rating scale; CVLT, California Verbal Learning Test; MMSE, Mini-Mental State Examination; NTB, Neurocognitive test Battery; TICS-M, The Modified Telephone Interview for Cognitive Status.

**Table 3 nutrients-12-03458-t003:** Summary of studies on vitamin A intake and AD.

Vitamins	Study Design	Participants [*n*]/Age [Year]	Assessment	Intervention/ Measurements	Duration	Results	References
MEASUREMENTS							
Vitamin A	cross-sectional study	333/≥60	MMSE, Dementia Rating Scale, Geriatric Depression Scale, Rey Auditory Verbal Learning	vitamin A serum level	1 measurement	(+) deficiency increases the risk of MCI	[30]
CONSUMPTION							
β-carotene	prospective study	49,493/average: 48	SCF, FFQ test	long-term intakes of carotenoids	22 years	(+) higher consumption of β-carotene is associated with a lower risk of cognitive performance	[31]
β-carotene	randomized double-blind, placebo-controlled study	2533/45–60	semantic and phonemic fluency tests, executive function	six 24-h dietary records	13 years	(−) consumption of fruit rich in β-carotene did not decrease the risk of developing cognitive impairment	[32]

(+)—confirmed or (−)—no confirmed association of intervention/concentration measurement/consumption with Alzheimer’s disease, FFQ, Food Frequency Questionnaire; MMSE, Mini-Mental State Examination; SCF, Subjective Cognitive Function; MCI, mild cognitive impairment.

**Table 4 nutrients-12-03458-t004:** Summary of studies on vitamin C and E consumption, supplementation, and blood measurements, and AD.

Vitamins	Study Design	Participants [*n*]/Age [Year]	Assessment	Intervention/Measurements	Duration	Results	References
INTERVENTIONS							
E	randomized controlled trial	613 with AD/ 78.8 ± 7.1	MMSE	2000 IU/day of α-tocopherol + 20 mg/day of memantine	2.27 years	(+) slowing functional decline	[38]
E	randomized controlled trial cohort study	7540/>60	MIS, TICS-m, NYU Paragraph Delayed Recall	400 IU of vitamin E/day and/or 200 μg of selenium/day	6 years	(−) supplementation does not prevent AD	[39]
C and E	randomized controlled trial	256/60–75	MMSE	300 mg of vitamin E + 400 mg of vitamin C/day	1 year	(−) supplementation does not prevent cognitive decline	[40]
MEASUREMENTS							
E	cross-sectional study	113/88.5 ± 6.0	MMSE	α- and γ-tocopherol brain levels	3 years	(+) vitamin E facilitates maintenance a presynaptic proteins level	[41]
E	cross-sectional study	716/>65	MMSE	tocopherols and total vitamin E levels	1 measurement	(+) lower levels of total tocopherols, total tocotrienols, and total vitamin E are associated with an increased likelihood of developing AD	[42]
CONSUMPTION							
C and E	double-blind, placebo-controlled	2533/45–60	verbal memory (RI-48 cued recall, semantic, and phonemic fluency tests) and executive function (trail-making and forward and backward digit span tests)	intake of fruit and vegetable intake (24-h dietary records)	13 years	(+) improvement of verbal memory	[32]
C	prospective, longitudinal cohort study	925/58–98	FFQ	intake of strawberries (vitamin C)	6.7 years	(+) possible reduction of AD	[43]

(+)—confirmed or (−)—no confirmed association of intervention/concentration measurement/consumption with Alzheimer’s disease, FFQ, Food Frequency Questionnaire; MIS, Memory Impairment Screen; MMSE, Mini-Mental State Examination; NYU, New York University; TICS-m, modified Telephone Interview for Cognitive Status.

**Table 5 nutrients-12-03458-t005:** Summary of studies on vitamin D supplementation and blood measurements in AD.

Vitamin	Study Design	Participants [*n*]/Age [Year]	Assessment	Intervention/ Measurement	Duration	Results	Reference
INTERVENTION							
D	randomized double-blinded placebo-controlled trial	4144/>65	WHIMS, MMSE	1000 mg of calcium carbonate + 400 IU of vitamin D(3)	8 years	(−) no association between treatment and incident cognitive impairment	[46]
MEASUREMENTS							
D	cross-sectional study	146/79.1 ± 7.0	MMSE	25(OH)D serum level	1 measurement	(+) higher serum 25(OH)D level are associated with MMSE score	[47]
D	cross-sectional study	208/79 ± 1	MMSE, CDR	serum 25-hidroxyvitamin D (25(OH)D) levels	1 year	(+) low vitamin D are associated with cognitive decline	[48]
D	randomized controlled trial	1154/79.7 ± 8.4	MMSE	serum levels of 25(OH)D	8 years	(+) low levels are associated with cognitive decline	[49]
D	cohort study	2990/76.5 ± 3.9	MMSE-KC, Test Digit Span	25(OH)D serum level	2 years	(−) no relationship between vitamin D levels and cognition	[50]
D	cohort study	661/≥65	3MS	Plasma 25-hydroxyvitamin D	10 years	(+) higher 25(OH)D concentrations increase risk of dementia and AD in women	[51]
D	retrospective, longitudinal cohort study	299/61 ± 10, 66 ± 8	MMSE	28 nutritional biomarkers in blood and 5 in cerebrospinal fluid, including 1.25(OH)2	2–5 years	(+) high vitamin D is associated with cognitive decline	[52]

(+)—confirmed or (−)—no confirmed association of intervention/concentration measurement/consumption with Alzheimer’s disease, 3MS, Modified Mini-Mental State; CDR, Clinical Dementia Rating scale; MMSE, Mini-Mental State Examination; MMSE-KC, Mini-Mental State Examination in the Korean version; WHIMS, Women’s Health Initiative Memory Study.

**Table 6 nutrients-12-03458-t006:** A review of meta-analyses and systematic reviews of recent years on the effects of nutrition on AD.

Factors	Type of Analysis	Results	Year	References
VITAMINS				
Vitamins B6, B12, and folate	systematic review and meta-analysis: 8 cross-sectional, 13 longitudinal	No link between the concentration of vitamin B in the blood and the risk of cognitive disorders.	2020	[66]
Vitamins B	meta-analysis: 11 large studies	Lowering homocysteine levels through vitamin B supplementation does not improve cognitive function.	2016	[67]
Folic acid, vitamin B12	meta-analysis: 68 studies	High homocysteine levels and low levels of folic acid and vitamin B12 increase the risk of AD.	2015	[56]
Vitamins C, E, and β-carotene	meta-analysis: 7 studies	Consumption of vitamins E, C, and β-carotene reduces the risk of AD. Vitamin E is the strongest predictor.	2012	[68]
Vitamin D	meta-analysis: 12 prospective cohort studies, 4 cross-sectional studies	Vitamin D deficiency increases the risk of dementia and AD.	2019	[69]
Vitamin D	meta-analysis: 6 prospective cohort studies	Serum vitamin D deficiency is associated with the risk of AD.	2019	[70]
Vitamin D	meta-analysis: 7 prospective cohort studies, 1 retrospective cohort study	Higher levels of 25(OH)D reduce the risk of AD.	2019	[71]
Vitamin E	meta-analysis: 31 observational studies and clinical trial	Lower *α*-tocopherol levels increase the risk of AD and MCI.	2019	[72]
Vitamin E	meta-analysis: 17 studies	Lower vitamin E levels are associated with the risk of AD.	2018	[73]
OTHER NUTRITIONAL FACTORS				
Fatty acids, foods rich in polyphenols	systematic review: 10 randomized clinical trials	Fatty acids and products rich in flavonoid are a protective factor in the development of dementia	2020	[64]
Ketogenic diet	systematic review: 10 randomized clinical trials	MCT and other ketogenic therapies improve cognition	2020	[59]
Meat consumption	systematic review and meta-analysis (PRISMA): 10 studies	No association between meat consumption and cognitive impairment	2020	[57]
Mediterranean diet	systematic review: 7 randomized clinical trials, 38 longitudinal studies	The diet is a protective factor to cognitive decline	2020	[58]
Nutritional formulas, fatty acid, ginseng, inositol, probiotics	systematic review: 32 randomized clinical trials	Interventions protect or delay the progression of cognitive decline	2020	[63]
Supplementation (vitamins B, vitamins E, omega-3 fatty acids, nutritional formulas)	systematic review and meta-analyses (PRISMA): 4 randomized clinical trials	No significant link between supplementation and cognitive function	2020	[65]
Traditional, healthy diet	systematic review and meta-analysis: 12 studies	Healthy diet reduces the risk of dementia	2020	[60]
Pro-inflammatory diet	systematic review and meta-analysis (PRISMA): 10 randomized clinical trials	Inflammatory diet is associated with cognitive decline	2019	[62]
Traditional, healthy diet and high-fat, high-glycemic diet	systematic reviews and meta-analyses (PRISMA): 26 studies	Healthy diet protects against the development of AD, unhealthy diet increases neurodegeneration	2019	[61]

MCT, Medium Chain Trigliceryde.

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
