# Peer review of "Vitamins in Alzheimer’s Disease—Review of the Latest Reports"

_nutrients, 2020, doi:10.3390/nu12113458_

Round 1

Reviewer 1 Report

The manuscript is a narrative review on the use of vitamins for patients with AD.

In table 2, the design is not clear for all studies. For example, a “3 centers study” is cross-sectional. These should be corrected accordingly. In parallel, Table 2 is highly heterogenous. The title refers to “Summary of studies on vitamin B intake and AD”, however, not all studies involve vitamin B intake, as some aim in measuring serum B vitamin levels also. Please correct accordingly, by either removing the studies without intake/supplementation data, or by splitting the table to 2 different ones, one with studies measuring levels and one with intake studies. If the second is selected, then the search used to identify studies is not adequate, as serum levels are not included in the search and this should be revised accordingly.

The same goes for Table 3, 4 and 5.

In Table 6, systematic reviews must also be included, especially given that the meta-analyses included do not concern intervention studies and RCTs, therefore not being able to provide causal evidence. Please include all SRs and MAs on nutrition, diet, supplementation and AD accordingly, otherwise the review is not up to date and not of high evidence.

I am adding some recent SRs for you:

10.1016/j.jamda.2020.08.020

10.1590/1980-57642020dn14-030008

10.1093/advances/nmaa073

10.3233/JAD-200499

10.1007/s10072-019-03976-3

Reviewer 2 Report

This is a nice, comprehensive review summarizing recent studies on the usage of different dietary vitamin supplements to treat AD progression. 

I would have liked to see a more fleshed out conclusion and discussion. How is the field moving? What are the future studies being pursued by the field? What kind of public health measures are being put in place by governments (such as Poland) to promote healthy aging? 

Instead of commenting on every single study with one or two sentences, I would have liked to see more of a summary that discusses each Vitamin supplementation collectively. Is there a consensus as to what works and what does not work? Were there any caveats as to why certain studies did not see large effects?

There were a few minor spelling mistakes and the article should be professionally edited for style and flow. 
